# Effects of an mHealth intervention for community health workers on maternal and child nutrition and health service delivery in India: protocol for a quasi-experimental mixed-methods evaluation

Sneha Nimmagadda,[1] Lakshmi Gopalakrishnan,[1] Rasmi Avula,[2] Diva Dhar,[3] Nadia Diamond-Smith,[4] Lia Fernald,[5] Anoop Jain,[5] Sneha Mani,[2] Purnima Menon,[2] Phuong Hong Nguyen,[6] Hannah Park,[4] Sumeet R Patil,[1] Prakarsh Singh,[7] Dilys Walker[4]

For numbered affiliations see end of article.

**Correspondence to**
Dr Sumeet R Patil;
srpatil@neerman.org

## ABSTRACT

**Introduction** Millions of children in India still suffer from poor health and under-nutrition, despite substantial improvement over decades of public health programmes. The Anganwadi centres under the Integrated Child Development Scheme (ICDS) provide a range of health and nutrition services to pregnant women, children <6 years and their mothers. However, major gaps exist in ICDS service delivery. The government is currently strengthening ICDS through an mHealth intervention called Common Application Software (ICDS-CAS) installed on smart phones, with accompanying multilevel data dashboards. This system is intended to be a job aid for frontline workers, supervisors and managers, aims to ensure better service delivery and supervision, and enable real-time monitoring and data-based decision-making. However, there is little to no evidence on the effectiveness of such large-scale mHealth interventions integrated with public health programmes in resource-constrained settings on the service delivery and subsequent health and nutrition outcomes.

**Methods and analysis** This study uses a village-matched controlled design with repeated cross-sectional surveys to evaluate whether ICDS-CAS can enable more timely and appropriate services to pregnant women, children <12 months and their mothers, compared with the standard ICDS programme. The study will recruit approximately 1500 Anganwadi workers and 6000+ mother-child dyads from 400+ matched-pair villages in Bihar and Madhya Pradesh. The primary outcomes are the proportion of beneficiaries receiving (a) adequate number of home visits and (b) appropriate level of counselling by the Anganwadi workers. Secondary outcomes are related to improvements in other ICDS services, and knowledge and practices of the Anganwadi workers and beneficiaries.

**Ethics and dissemination** Ethical oversight is provided by the Committee for the Protection of Human Subjects at the University of California at Berkeley, and the Suraksha Independent Ethics Committee in India. The results will be published in peer-reviewed journals and analysis data will be made public.

### Strengths and limitations of this study

► The study can provide important evidence on whether and the extent to which a large-scale mHealth intervention can improve maternal and child health and nutrition service delivery by the world's largest child development programme in India.

► Application of the gold-standard cluster-randomised controlled trial design was not possible because of pre-determined programme assignment and rapid roll-out of the programme. Therefore, to find attributable impacts, this evaluation settles for a scientifically less robust but practicable quasi-experimental design consisting of matched control-villages and repeated cross-sectional measurements.

► Measurement biases may exist because blinding is not possible and primary outcomes are measured subjectively via interview-based recall or observations.

► Higher order impacts may be underestimated as the follow-up period of <12 months may be too short for the intervention to stabilise.

**Trial registration number** ISRCTN83902145

## INTRODUCTION
### Background and problem context

Millions of children in India continue to suffer from poor health and under-nutrition, despite decades of government programmes aimed at reducing this burden and some impressive gains through these years. In 2015–2016, 36% of children under 5 years of age were underweight, 38% were stunted and 21% were wasted as per the National Family Health Survey (NFHS-4), and these numbers represent only modest improvements over the past decade. Micronutrient deficiencies

are widespread, with more than 58% of preschool children suffering from iron deficiency anaemia. Infant and neonatal mortality rates also remain high at 41 and 30 per 1000 live births respectively, despite substantive reductions over past decades[1]

The Integrated Child Development Services Scheme (ICDS), launched in 1975, is one of India's national flagship programmes to support the health, nutrition, and development needs of children below 6 years of age and pregnant and lactating women, through a network of Anganwadi Centres (AWCs), each typically serving a population of 800–1000.[2 3] Early observational studies found that ICDS is associated with better coverage and delivery of services related to nutrition, healthcare, and pre-school education and improved maternal and child nutrition.[4–6] Using the NFHS data from 2005 to 2006, Kandpal[7] and Jain[8] found that ICDS is associated with small to modest improvements in child health and nutrition, especially among the most vulnerable populations.

However, several reviews and evaluations of ICDS over the past 18 years have also found persistent gaps, including inadequate infrastructure at the AWC, Anganwadi worker (AWW) service delivery issues (eg, poor quality supplementary food, few home visits and no counselling etc), human resource issues (eg, vacancies, increasing range of duties expected of the AWWs, inadequate training of AWWs, limited supervision etc) and poor data management (eg, irregularities in record keeping at AWCs, ineffective monitoring of service delivery etc).[3 4 9–11] The most recent NFHS (2015–206) also highlights the gaps in ICDS service delivery. Only about 59% of children under 6 years received any service from an AWC, 53% received supplementary food services and 47% were weighed. Similarly, only 60% of mothers received any AWC services during pregnancy, and 54% received any service during the breastfeeding phase.[1]

With a goal to improve the functioning of ICDS, the Government of India launched the ICDS Systems Strengthening and Nutrition Improvement Programme (ISSNIP) in 2012 which focused on infrastructure upgradation and training of AWWs to build their knowledge on health and nutrition topics under the Incremental Learning Approach. At the same time, a pilot-scale mHealth intervention to improve ICDS service delivery was implemented in Bihar between 2012 and 2013. A randomised controlled trial of this intervention found a significant increase in the proportion of beneficiaries receiving visits from frontline workers at different lifestages - last trimester of pregnancy (42% vs 52%), first week after delivery (60% vs 73%) and complementary feeding stage >5 months after delivery (36% vs 45%).[12] The intervention also significantly increased the proportion of beneficiaries receiving at least three antenatal care visits (29% vs 50%), the proportion of beneficiaries consuming at least 90 iron folic acid tablets during pregnancy (11% vs 17%), the proportion of mothers breastfeeding immediately after birth (62% vs 76%) and the proportion of mothers starting complementary feeding at the right time (32% vs 41%). Subsequently, the ISSNIP was restructured in 2015 by integrating ICDS in seven states with an at-scale mHealth intervention called Common Application Software (ICDS-CAS) installed on smart phones and with accompanying multilevel data dashboards. This system is intended to be a job aid for frontline workers, supervisors and managers, and aims to ensure better service delivery and supervision by enabling real-time monitoring and data-based decision-making.

While there is a growing body of evidence on the effectiveness of mHealth interventions, it almost entirely consists of small-scale studies or pilot interventions under well controlled settings, and often of poor research quality. For example, a systematic review examining 17 studies set in low and middle-income countries found that small scale mHealth interventions, particularly those delivered using SMS, were associated with increased utilisation of healthcare, including uptake of recommended prenatal and postnatal care consultation, skilled birth attendance and vaccination, but only two of these studies were graded as being at *low risk of bias*.[13] Barnett and Gallegos[14] reviewed nine studies that assessed the impact of using of mobile phones for health and nutrition surveillance, and found that while the available evidence suggests that mobile phones may play an important role in nutrition surveillance by reducing the time required to collect data and by enhancing data quality, the available evidence is of poor methodological quality and is generally based on small pilot studies and mainly focuses on feasibility issues. Another recent systematic review of 25 studies found evidence that mobile tools helped community health workers improve the quality of care provided, the efficiency of services and the capacity for programme monitoring.[15] However, most of these studies were pilots and provided little or no information about the effectiveness of mHealth interventions when integrated with large-scale public health programmes.

This study seeks to address this critical gap in the evidence base in the context of the largest public health and nutrition programme in the world, ICDS, with 1.4 million AWCs serving at the grassroots level across India. The impact evaluation is conducted in two large states in India — Madhya Pradesh (MP) and Bihar — using a quasi-experimental, matched-controlled pre-measurement and post-measurement design. The overall evaluation framework consists of additional components such a process evaluation, a technology evaluation and an economic evaluation.

This evaluation is also timely as India launched the National Nutrition Mission on 8 March 2018 with the goal of reducing malnutrition in a phased manner across entire of India and subsumed ISSNIP and ICDS-CAS under it.[16] Therefore, ISSNIP and ICDS-CAS are poised to be scaled-up rapidly to reach almost the entire population of India through 1.4 million AWCs by 2020. This scale-up effort will be informed by robust evidence on the effectiveness of the mHealth intervention, as well as on how its implementation can be improved.

## The ICDS-CAS intervention

Currently, the ICDS-CAS intervention is being implemented at scale in seven states, covering over 107 000 AWCs, and through them, a population of 9.8 million registered beneficiaries. The intervention consists of two components as follows.[17]

(1) An android CAS application and smartphones for AWWs and the female supervisor: The CAS app was developed on an open source mobile platform (CommCare). The app digitises and automates ten of the eleven ICDS paper registers maintained by AWWs, enables name-based tracking of beneficiaries, prioritises home visits at critical life-stages through a home visit-scheduler, improves record keeping and retrieval of growth and nutrition status of children, helps track immunisation, monitors the timeliness and quality of different services delivered by AWWs, and includes checklists and videos as job aids. A female supervisor typically manages a cluster of 10–20 AWCs and the CAS app is expected to help her monitor AWWs remotely, assess quality of service delivery, and serves as a job aid to train AWWs. The app is installed on new smartphones that are provided to the AWWs and supervisors. Both AWWs and supervisors are trained on the use of the app and how the features help them improve service delivery. Helpdesks at block and district levels for technical support are also established.

The CAS app is especially expected to improve home visit service delivery by AWWs through improved channels of information (easy access to past records of the beneficiary for customised messaging, educational animation videos as a job aid, life-stage-appropriate checklists for counselling messages) and timely nudges (automatic creation of visit-due lists, alerts for approaching or missed visits and timely intimation of delays to the female supervisor). Thus, improved home visits in terms of timeliness, frequency, and perhaps, a more effective message delivery mechanism are expected to result in increased knowledge and better recall of correct health and nutrition practices by the beneficiaries and higher demand for related government services. However, for the actual behaviours to change and sustain, supply side constraints must be addressed to meet the demand for services (eg, adequate supply of supplementary food, adequate provisions of Iron Folic Acid (IFA) tablets, regular immunisation camps, etc). Such improvements can be expected only in the mid-to-long-term because they are beyond the sphere of influence of ICDS-CAS and need more ICDS-wide improvements.

(2) A web-enabled dashboard for real-time monitoring by ICDS officials: Data generated at the AWC-level are aggregated and analysed via web-enabled dashboards for Child Development Project Officers at the project-level (typically an administrative block with 80–100 AWCs), District Programme Officers, the state ICDS Directorate and the Ministry of Women and Child Development (MWCD) at the national level. For example, the monthly progress reports are prepared manually at the AWC-level and then aggregated to the project-level which require

weeks to be finalised and reviewed, but the CAS app and dashboards will automate and produce these reports in almost real-time. The dashboard infographics are expected to help identify bottlenecks at various levels more efficiently, help prioritise local issues, and allow managers to take data-driven decisions.

Figure 1 presents the ICDS-CAS information flow from the AWC through to the MWCD. The logic model in figure 2 summarises how ICDS-CAS is expected to improve service delivery, and ultimately improve health and nutritional outcomes in mothers and their children. The listed short-term outcomes are those expected to be achieved in the planned evaluation follow-up period of <12 months. The longer-term outcomes related to improved health and nutrition will be measured (except improvements in cognitive abilities) and analysed from a learning perspective, but these remain aspirational in the context of this evaluation study.

## Evaluation framework and research objectives

The ICDS-CAS evaluation framework consists of four components – an impact evaluation, a process evaluation, an economic evaluation and a technology evaluation. This paper describes the protocol for the impact evaluation in detail to guide the final analysis plan. The other three components are summarised online in supplementary figure 1 without binding details because their objectives, scope and methods can change as per the evolving learning needs of policy makers. We intend to analyse and publish impact evaluation and process evaluation as standalone research articles in peer-reviewed journals. The analysis and publication plan for the other components is yet to be determined.

Corresponding to the short-term strategic objective of ICDS-CAS to improve quality and quantity of AWW and beneficiary interactions, the main research questions that impact evaluation seeks to answer are:

1. Does ICDS-CAS improve the timeliness or frequency of home visits by AWWs for pregnant women, infants and their mothers?
2. Does ICDS-CAS improve the extent or level of counselling by AWWs to pregnant women and mothers of infants?

## METHODS AND ANALYSIS
### Study setting

The ICDS-CAS programme is being implemented in 57 districts from seven ISSNIP states in India where the burden of under-nutrition is highest. This evaluation is restricted to two states, MP and Bihar, which were selected because of the possibility of selecting an ISSNIP district as a control, willingness of the states to support the evaluation, and the suggestions by the MWCD and the funding agency. Online supplementary table 1 presents key health and nutrition related indicators for villages in MP and Bihar. Both states have a high burden of under-five mortality (69 per 1000 live births in MP and 60 in

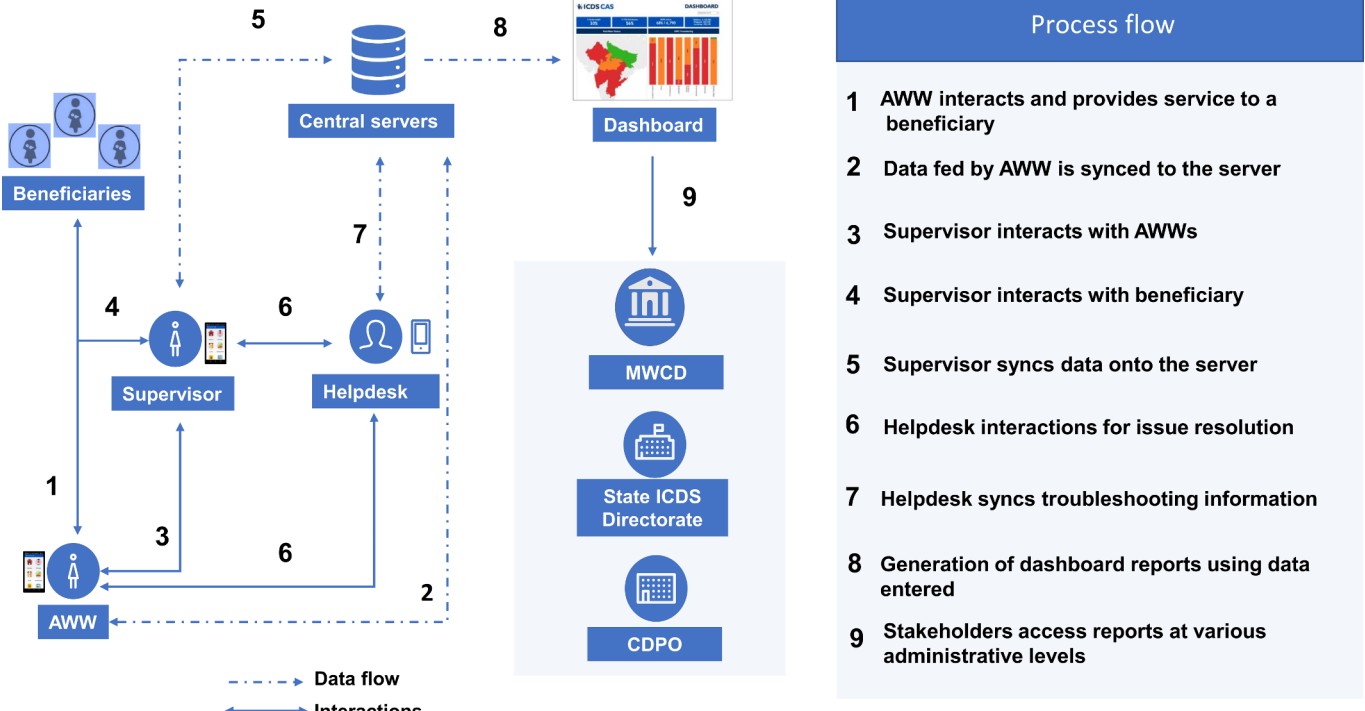

**Figure 1** ICDS-CAS information flow from the Anganwadi Centre to the Ministry of Women and Child Development. Solid lines correspond to interactions. Dotted lines correspond to data flow. AWW, Anganwadi worker; CDPO, Child Development Project Officer; ICDS-CAS, Integrated Child Development Services-Common Application Software; MWCD, Ministry of Women and Child Development.

Bihar), stunting (43.6% of children aged 0–5 years in MP, and 49.3% in Bihar) and anaemia (>55% of children and pregnant women in both states). Antenatal and delivery-related indicators are, in general, worse in Bihar, whereas MP has relatively poor demographic, water-sanitation, education and mortality related indicators– 8.3% of mothers in MP and 3% in Bihar had full antenatal care; 79.5% of households in MP and 98.2% in Bihar had an improved drinking-water source; and 50.2% of children aged 12–23 months in MP and 61.9% in Bihar were fully immunised.

### Overview of the identification strategy

The attributable effects of ICDS-CAS will be identified using a quasi-experimental matched, controlled design with repeated cross-sectional pre-intervention and post-intervention measurements. This identification strategy is grounded in the Neyman–Rubin potential outcomes model where a matched cohort design can yield unbiased estimates of the causal effects under a strong assumption that all confounders are measured and balanced between the intervention and comparison groups.[18–20] We use a 1:1 nearest neighbour propensity score matching (PSM) method to identify pairs of intervention and comparison villages.[21–25] We plan to identify the effects of ICDS-CAS by comparing post-intervention outcome indicators between the matched groups, while controlling for any pre-intervention or baseline differences in the outcomes averaged at the village level and adjusting for the matched paired design.

### Sample design and power calculations

The initial sample size was determined as 400 villages in each arm with one AWW and, on average, three mother-child dyads in each village to measure a relative effect of 15% in a standardised counterfactual outcome [Normal(0,1)], with a significance level of 0.05, power of 80% and intra-cluster correlation (ICC) of 0.15. The actual baseline survey sample consisted of 852 villages to account for refusals and loss to follow-up.

After the baseline survey was conducted in June-August 2017, completely separate from our efforts to design the evaluation, the Indian government decided to include ICDS-CAS in the National Nutrition Mission. Consequently, the original evaluation objectives were revised to estimate the effects of ICDS-CAS separately for MP and Bihar to draw deeper insights into the heterogeneity of impacts. Therefore, the evaluation sample needed to be powered to detect smaller magnitudes of effects on a few process-related secondary outcomes. We plan to increase the sample power by following the same panel of villages but recruiting almost twice as many participants in each village – up to two AWWs and up to eight mother-child dyads. Assuming 200 pairs of villages and 1200 mothers per arm in each state, the sample will have 80% power at a significance level of 0.05 to detect an absolute difference of 5%–9%-points from the counterfactual levels between 10%–50% with ICC between 0.15–0.30. With 400 AWWs per arm in each state, the sample will have 80% power to detect effects of 8–12%-points for AWW level outcomes

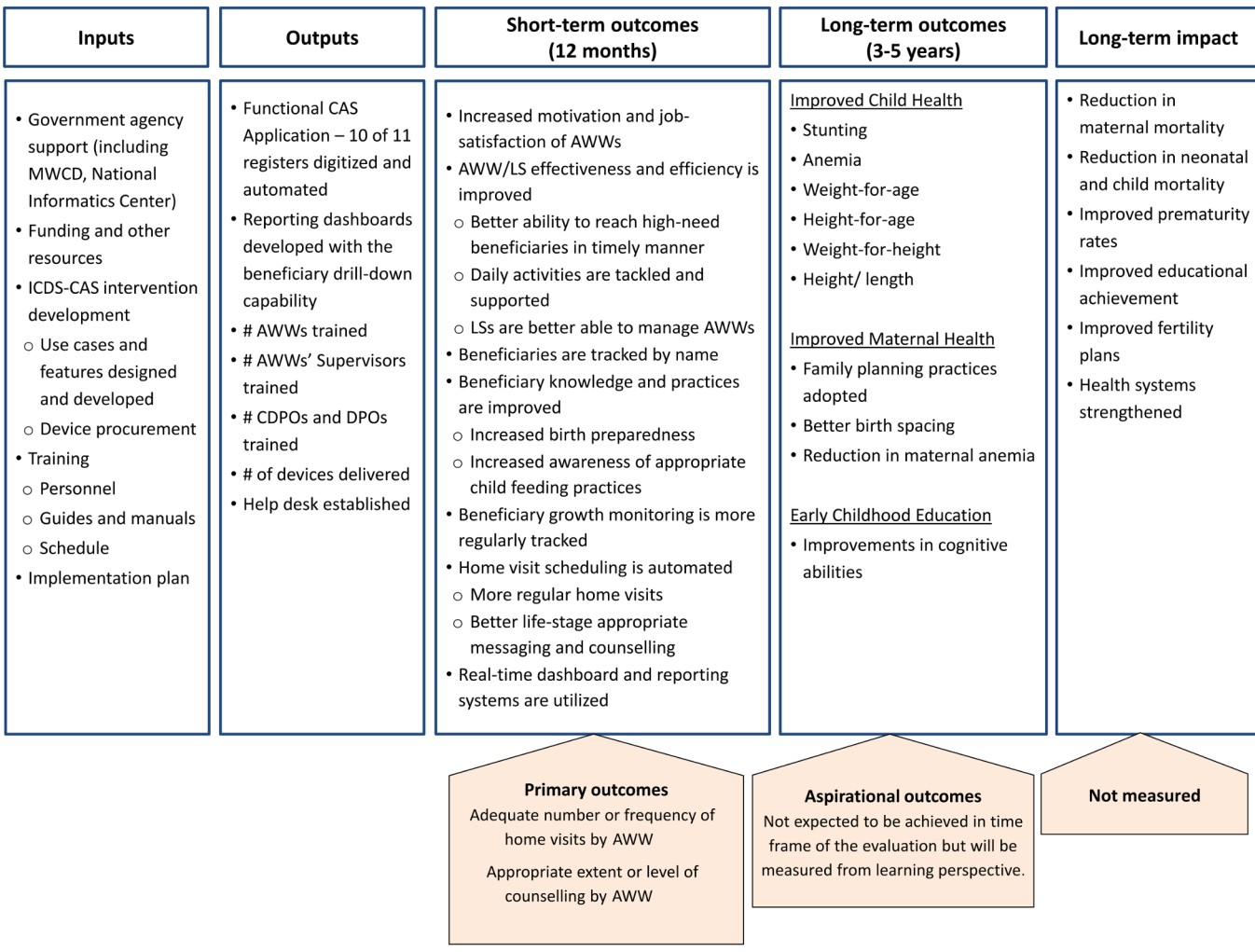

| Inputs | Outputs | Short-term outcomes (12 months) | Long-term outcomes (3-5 years) | Long-term impact |
|---|---|---|---|---|
| • Government agency support (including MWCD, National Informatics Center)<br>• Funding and other resources<br>• ICDS-CAS intervention development<br> ○ Use cases and features designed and developed<br> ○ Device procurement<br>• Training<br> ○ Personnel<br> ○ Guides and manuals<br> ○ Schedule<br>• Implementation plan | • Functional CAS Application – 10 of 11 registers digitized and automated<br>• Reporting dashboards developed with the beneficiary drill-down capability<br>• # AWWs trained<br>• # AWWs' Supervisors trained<br>• # CDPOs and DPOs trained<br>• # of devices delivered<br>• Help desk established | • Increased motivation and job-satisfaction of AWWs<br>• AWW/LS effectiveness and efficiency is improved<br> ○ Better ability to reach high-need beneficiaries in timely manner<br> ○ Daily activities are tackled and supported<br> ○ LSs are better able to manage AWWs<br>• Beneficiaries are tracked by name<br>• Beneficiary knowledge and practices are improved<br> ○ Increased birth preparedness<br> ○ Increased awareness of appropriate child feeding practices<br>• Beneficiary growth monitoring is more regularly tracked<br>• Home visit scheduling is automated<br> ○ More regular home visits<br> ○ Better life-stage appropriate messaging and counselling<br>• Real-time dashboard and reporting systems are utilized | <u>Improved Child Health</u><br>• Stunting<br>• Anemia<br>• Weight-for-age<br>• Height-for-age<br>• Weight-for-height<br>• Height/ length<br><br><u>Improved Maternal Health</u><br>• Family planning practices adopted<br>• Better birth spacing<br>• Reduction in maternal anemia<br><br><u>Early Childhood Education</u><br>• Improvements in cognitive abilities | • Reduction in maternal mortality<br>• Reduction in neonatal and child mortality<br>• Improved prematurity rates<br>• Improved educational achievement<br>• Improved fertility plans<br>• Health systems strengthened |

**Primary outcomes**
Adequate number or frequency of home visits by AWW
Appropriate extent or level of counselling by AWW

**Aspirational outcomes**
Not expected to be achieved in time frame of the evaluation but will be measured from learning perspective.

**Not measured**

**Figure 2** Logic model of ICDS-CAS and measurement of outcomes. AWW, Anganwadi worker; CDPO, Child Development Project Officer; DPO, District Programme Officer; ICDS-CAS, Integrated Child Development Services -Common Application Software; LS, lady supervisor; MWCD, Ministry of Women and Child Development.

from the counterfactual levels of 10%–40% with ICC between 0.25–0.45.

### Sampling and recruitment of study participants

Figure 3 summarises the sampling and recruitment of study participants for the baseline survey. First, we sampled intervention districts and selected geographically and administratively matched comparison districts. Then, we randomly sampled intervention villages in two steps, first sampling the blocks and then the villages. Finally, we pair-matched intervention villages with villages from the comparison districts and selected the best matched 426 pairs as discussed next.

*Sampling of Intervention and Comparison Districts*: We randomly sampled three pairs of geographically and administratively matched intervention and comparison districts which shared a boundary and belonged to the same ICDS division to control for division-level confounders related to ICDS administration and management as well as cultural, social, environmental and economic factors at the micro-region scale. Within each

state, we first sample three intervention districts. The corresponding control districts were selected by default if only one eligible district was available within the division, or randomly, if multiple comparison districts were available. In MP, the sample purposively included one pair of *tribal-only* districts from an equity-focused evaluation perspective. Figure 4 depicts the states and districts included in the evaluation.

*Selecting Matched Pairs of Treatment and Control Villages*: From each selected district, we randomly sampled two blocks in MP and three blocks in Bihar. Next, we randomly sampled 345 villages from six intervention blocks in MP and 315 villages from nine ICDS-CAS blocks in Bihar. In both states, the sample frame was restricted to villages with a population of 500 or more to increase the possibility that the villages are sufficiently large to be served by dedicated AWCs. Next, we matched the sampled treatment villages with villages from the paired comparison district using a 1:1 nearest neighbour PSM method using the following variables from Census 2011 for matching:

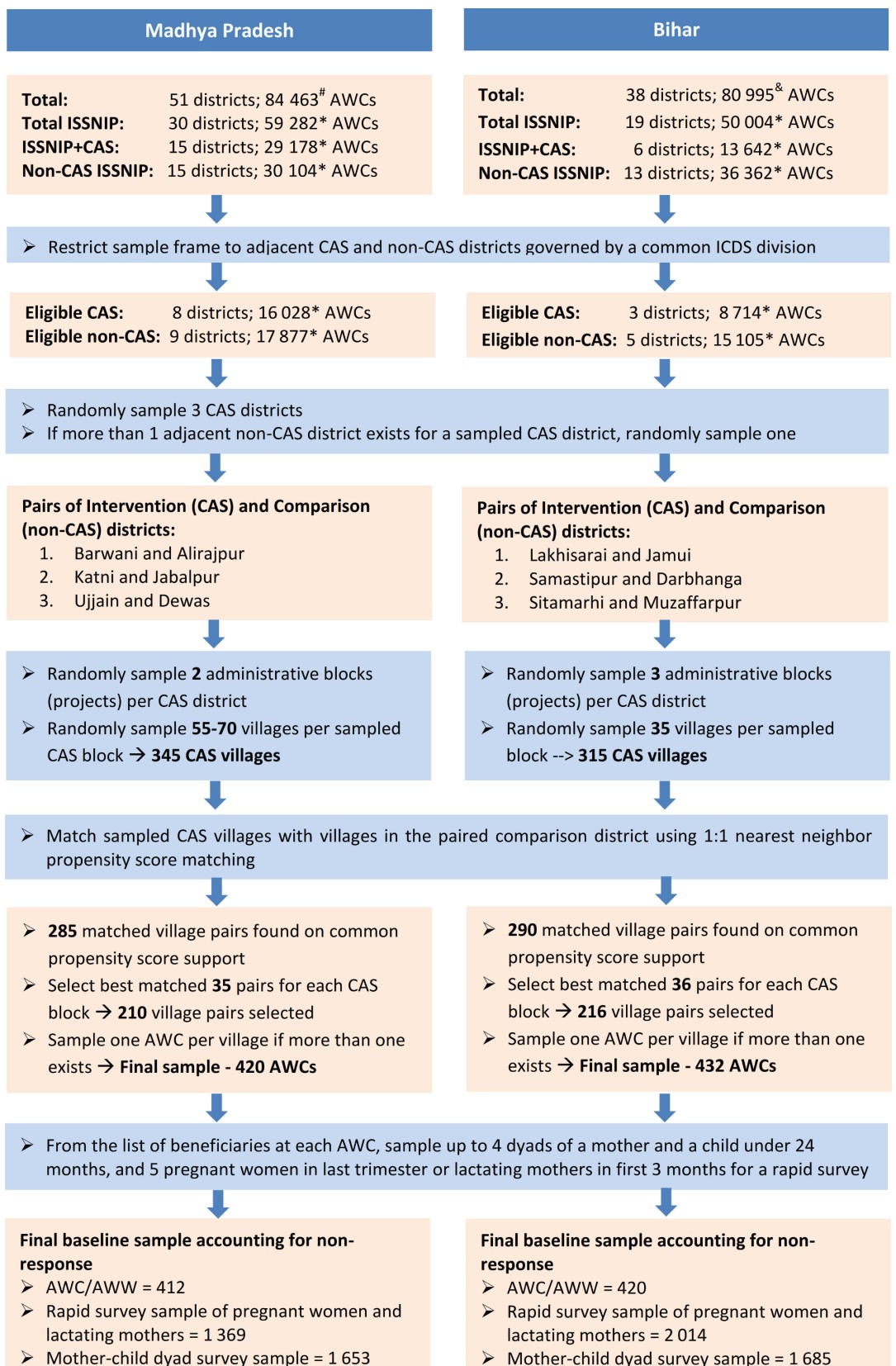

**Figure 3** Sampling of study participants from Madhya Pradesh and Bihar. AWC, Anganwadi Centre; AWW, Anganwadi worker; CAS, Common Application Software; ICDS, Integrated Child Development Services; ISSNIP, ICDS Systems Strengthening and Nutrition Improvement Programme. Sources: # Women and Child Development Department, Government of Madhya Pradesh (MIS) http://mpwcdmis.gov.in/ (See1m5bxnuzmixun00bsctvfj))/DataEntryAwc.aspx (Accessed 8 June 2018); & Integrated Child Development Services, Government of Bihar http://www.icdsbih.gov.in/AnganwadiCenters.aspx?GL=16 (Accessed 8 June 2018); *Programme documentation from implementing agencies.

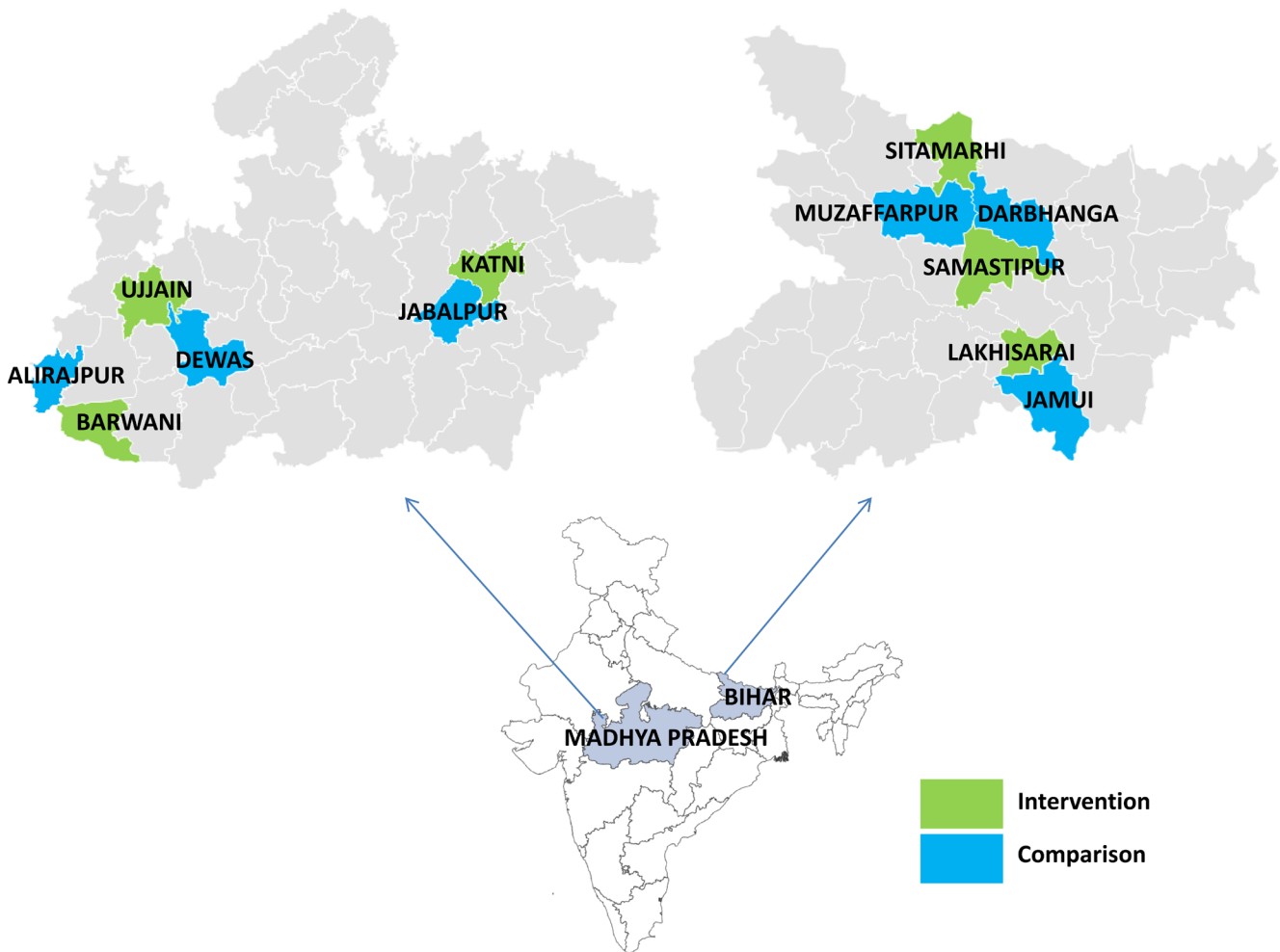

**Figure 4** Sampled intervention and comparison districts. Green areas indicate intervention districts. Blue areas indicate comparison districts.

distance between village and block headquarters (kilometres); population; % of schedule caste or schedule tribe households; if a village is served by public transport; if a village is connected to a major road; if a village has a public ration shop; if a village has a post office; if a village has a bank; % of households in a village with a bank account; if a village has an agricultural society; if a villages has a self-help group; % of households in a village serviced by closed drainage system; % of households in a village with improved source of drinking water; % of households in a village with improved sanitation facility; % of households in a village using electricity as the main source of light; % of households in a village with a *pucca* house; household asset index for the village).[20–23]

| Table 1 | Comparison of matching performance in reducing bias | | | | | |
|---|---|---|---|---|---|---|
| | **Madhya Pradesh** | | | **Bihar** | | |
| | **Ujjain-Dewas** | **Barwani-Alirajpur** | **Katni-Jabalpur** | **Samastipur-Darbhanga** | **Lakhisarai-Muzaffarpur** | **Sitamarhi-Jamui** |
| Standardised mean bias - before matching | 16.6 | 27.5 | 40.3 | 27 | 35.4 | 40.8 |
| Standardised mean bias - after matching | 6.5 | 8 | 9.9 | 8.3 | 6.6 | 9.7 |
| % reduction in mean bias | 61% | 71% | 75% | 69% | 81% | 76% |
| P-value of LR test - before matching | 0.000 | 0.000 | 0.000 | 0.000 | 0.000 | 0.000 |
| P-value of LR test - after matching | 0.929 | 0.929 | 0.905 | 0.788 | 0.997 | 0.490 |
| Mean difference in propensity score | 0.002 | 0.072 | 0.131 | 0.004 | 0.046 | 0.181 |

LR, log reduction.

Table 1 summarises the matching performance in terms of reduction in the sum of standardised mean bias after matching. The standardised mean bias was reduced by 61%–81% after matching across the six district pairs. The Log Reduction test statistic before matching was statistically significant (p<0.001), but became insignificant after matching (p>0.75), suggesting that both groups are statistically similar on average after the matching procedure. We then selected the best matched village pairs– in terms of the difference in the propensity scores of the matched pairs –35 pairs of villages for each intervention block in MP and 24 pairs of villages for each intervention block in Bihar. Overall, we selected 216 matched pairs (432 villages) in Bihar and 210 matched pairs (420 villages) in MP.

*Recruitment of Study Participants*: In the already conducted baseline survey, we randomly sampled one AWC if more than one AWC existed in a selected village. Using the list of beneficiaries available at the selected AWC as a sample frame, we randomly sampled four mother-child dyads after stratification by life-stages: (1) mother of child aged <3 months, (2) mother of child aged three to <6 months, (3) mother of child aged six to <12 months and (4) mother of child aged 12 to <24 months. Additionally, we randomly sampled up to five pregnant women or mothers with a child <3 months for more in-depth analysis of ICDS services during pregnancy and child birth. To measure AWC-level outcomes, we conduct a survey of the selected AWWs as well.

The endline will be a repeated cross-sectional sample where pregnant women and mothers with children <12 months will be recruited using the list of beneficiaries available at the AWC at that time. We do not plan to recruit children older than 12 months at the endline because the ICDS-CAS intervention would be active for <12 months and we can realistically measure the changes only among children who were born after the ICDS-CAS implementation started.

All survey participants are/will be recruited for the study after being administered informed consent as per the Institutional Review Board approved protocol. Additional assent is taken just before taking anthropometric measurements (height and weight) of the children.

### Patient and public involvement

This research was done without patient or public involvement in conceptualisation, design, implementation, analysis, manuscript development or dissemination.

### Primary outcomes

The primary outcomes to assess the effectiveness of ICDS-CAS compared with the standard ISSNIP and ICDS are:

1. The proportion of pregnant women and mothers of children <12 months who received adequate number of home visits by AWWs in the past 3 months (adequate number will be the minimum number of visits a respondent must receive as per ICDS guidelines for

the current life-stage/age. Additionally, we will use, as a supporting indicator, the number of visits as a continuous outcome indicator.[26]); and

2. The proportion of pregnant women and mothers of children <12 months who received appropriate extent or level of counselling from AWWs during their interactions (at home, at AWCs or in other settings) in the past 3 months (appropriate level of counselling will be a recall of at least half of the correct messages/counselling that a respondent should receive as per ICDS guidelines for the current life-stage/age. Additionally, we will use, as a supporting indicator, number of correct messages or services recalled by the respondent as a continuous outcome.[26]).

### Secondary outcomes

Several outputs and outcomes according to the logic model presented in figure 2 are secondary outcomes in this evaluation study as discussed before. These include outcomes related to supervisory and capacity building support to AWWs, infrastructure and supplies at AWC, AWW level outcomes (motivation, satisfaction, knowledge, time allocation for services and record keeping, time allocation for service delivery, number of beneficiaries served) and additional ICDS services that can be improved by ICDS-CAS but also critically dependent on other external factors (growth monitoring of children, provision of IFA and supplemental nutrition, immunisation tracking, referrals, etc). We will also measure higher order but distal or aspirational outcomes related to knowledge, practices, health and nutrition at the beneficiary level.

### Outcome measurements

All beneficiary-level and AWW-level outcomes will be measured through structured interviews, with verification of registers and documents wherever possible. The household instruments were developed using standard questions from the WHO and Demographic and Health Survey frameworks, and capture information about the demographic and socio-economic characteristics of the household, birth history and family planning, pregnancy care, awareness and utilisation of AWC services at different life-stages, IYCF practices, immunisation, knowledge of health and nutrition and child health and nutritional outcomes (child growth). The AWW instrument captures information about coverage of beneficiaries, service delivery, supervisory support, work incentives and motivation, training received, time allocation, knowledge, and infrastructure and supplies at the AWC. The interviews are administered in *Hindi* on an android tablet-based SurveyCTO™ platform.

### Analysis plan

The effects of the ICDS-CAS intervention will be estimated as,

$$Y_{ij,\ t=1} = \beta_0 + \beta_1 . T_j + Pair\ ID_K + \overline{Y}_{j,t=0} + \varepsilon,$$

where

$Y_{ij,\ t=1}$ = an outcome of interest for beneficiary $i$ in village $j$ at endline ($t$=1);

$T_j$ = indicator variable which is 1 for intervention and 0 for comparison villages;

$Pair\ ID_K$ = fixed effects to account for $k$ pairs of matched villages;

$\overline{Y}_{j,t=0}$ = average pre-intervention or baseline (t = 0) level of the outcome Y in the village;

$\varepsilon$ = the error term of the model; and

$\beta_1$ = the effect of the intervention on outcome Y.

We will adjust for the pre-intervention average level of outcome in a village ($\overline{Y}_{j,t=0}$) to control for, *at least*, the measured or unmeasured *time invariant* village-level and higher-level confounders. To the extent that the endline questionnaires or sampling are different from those in the baseline, the construction of the outcome indicators in the baseline and endline may differ slightly, but such differences (if any) will not affect the interpretation of impact parameters estimated using the above model specification. Additionally, as a consistency check, we will estimate adjusted treatment effects that control for any observed imbalance in important baseline covariates. We may additionally control for covariates that can help increase the precision of impact estimates. We do not plan to correct the p-values for multiple comparisons because we only test a limited number of indicators for primary outcomes. However, for secondary outcomes, when multiple indicators related to a topic or theme are compared, we will adjust the p-values for multiple comparisons. We also plan to estimate the effects by adjusting for pairing only at the district level and not at the village-level if accounting for village matching results in substantial sample loss. All analyses will be done in STATA and replicated by two different analysts.

### Baseline balance

Online supplementary tables 2 and 3 present the baseline balance at AWC, household and individual beneficiary levels. The balance is assessed in terms of group mean difference after adjusting for the pairing of villages. As a robustness check, the group mean differences obtained by adjusting only for the district pairing (and not the village pairs) are also presented. Overall, practically perfect balance in terms of exogenous variables such as household or individual characteristics is achieved, but there are a few differences in the AWC and AWW level characteristics and service delivery. Almost all indicators related to home visits and growth monitoring were balanced as the magnitude of the group differences are of little practical significance, except for the growth monitoring related outcomes in Bihar. A few secondary outcomes were meaningfully different as well. However, such differences are expected when more than 100 covariates and outcome indicators are tested for balance. We also do not see a discernible pattern where control or intervention groups are consistently better or worse off

than the other group. Considering the preponderance of a highly similar distribution of variables, we infer that the matching resulted in exchangeable or balanced groups, and any residual confounding at the community level can be removed by controlling for the baseline outcomes.

### DISCUSSION

The evaluation will provide evidence on whether and to what extent ICDS-CAS mHealth can improve health and nutrition service delivery beyond what is feasible with traditional non-technology-based approaches under ISSNIP. Additionally, the analysis of a range of lower order outputs and outcomes can help us identify the pathways through which ICDS-CAS has worked, or the critical failure points.

The study faces a few limitations in identifying unbiased estimates of the programme effects due to the nature of the intervention and constraints on the study design. First, confounding or selection bias cannot be theoretically ruled out in an observational study such as ours. While the matching procedure appears to be successful, it may not have removed all residual and unobserved confounding. Measurement biases including the Hawthorne effect[27] are possible because the outcomes are measured through interview recalls and observations. The external validity of the findings can be questioned because the purposive sampling of states, pairing of districts and the PSM-based sampling of villages do not result in a statistically representative sample of entire ICDS-CAS programme area. Finally, as is the case with most large-scale programmes, ICDS-CAS implementation may be delayed and the planned follow-up period may not be adequate for the impacts to materialise.

While these limitations are common in observational studies, the study team has tried to minimise the risk to validity of the findings by reducing the observed pre-intervention imbalance using a large set of variables from Census 2011 for matching, controlling for at least the cluster level time-invariant confounders by using a repeated cross sectional design, measuring the primary outcomes at the beneficiary level (beneficiaries will be blinded to their intervention status in the study), measuring a large set of indicators as per the logic model to test whether ICDS-CAS is working through hypothesised pathways and delaying the endline survey as much as possible before the intervention is implemented in the control districts. The evaluation framework also includes other components which can assess the intervention using mixed-methods approaches, and can help build confidence in the study findings.

Overall, this study will contribute to the evidence base on whether mHealth interventions can improve community health worker efficiency and effectiveness. This is also a highly policy-relevant evaluation which can inform scale-up of the intervention to potentially cover the entire country by 2020.

## ETHICS AND DISSEMINATION

The results will be published in peer-reviewed journals and presented in conferences and dissemination meetings.

**Author affiliations**
[1]NEERMAN, Center for Causal Research and Impact Evaluation, Mumbai, India
[2]International Food Policy Research Institute, New Delhi, India
[3]Bill and Melinda Gates Foundation India, New Delhi, Delhi, India
[4]University of California San Fransisco, San Fransisco, USA
[5]School of Public Health, University of California, Berkeley, Berkeley, California, USA
[6]International Food Policy Research Institute, Washington, District of Columbia, USA
[7]Institute of Labour Economics (IZA), Seattle, Washington, USA

**Contributors** DW and LF are the principal investigators on the grant from Bill and Melinda Gates Foundation. ND-S, LF, PM, HP, SRP, PS, DW contributed to the study design and lead different sub-components of the study, DD commissioned the study and reviewed the study design, all authors were involved in protocol development and finalising instruments. SN and SP led the sampling. Data collection was mainly managed by SN, LG, RA, SM, AJ. SN, PHN, ND-S and SP conducted analyses and SN, LG, SP developed the first draft of the manuscript. All authors reviewed, revised and approved the final manuscript.

**Funding** This study is funded by Grant No. OPP1158231 from Bill and Melinda Gates Foundation to the University of California, San Francisco and University of California, Berkeley.

**Competing interests** DD is a Program Officer with the Measurement, Learning & Evaluation (MLE) team at the Bill & Melinda Gates Foundation (BMGF) India Country Office. BMGF has funded this study as well as the support to the scale-up of the ICDS-CAS programme. However, as part of the MLE team, DD has had no role in the ICDS-CAS program design or implementation and was responsible for conceptualizing and commissioning the evaluation. She continuous to advise on study design, analysis and communication of findings to stakeholders.

**Patient consent for publication** Not required.

**Ethics approval** Protocols have been reviewed and approved by institutional review boards at the University of California, Berkeley (Ref. No. 2016-08-9092), and the India-based Suraksha Independent Ethics Committee (Protocol No. 2016-08-9092).

**Provenance and peer review** Not commissioned; externally peer reviewed.

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
