## [Reviewer comments · BMJ Open]

ARTICLE DETAILS

TITLE (PROVISIONAL)	Effects of an mHealth intervention for community health workers on maternal and child nutrition and health service delivery in India: Study protocol for a quasi-experimental mixed methods evaluation
AUTHORS	Patil, Sumeet; Nimmagadda, Sneha; Gopalakrishnan, Lakshmi; Avula, Rasmi; Dhar, Diva; Diamond-Smith, Nadia; Fernald, Lia; Jain, Anoop; Mani, Sneha; Menon, Purnima; Nguyen, Phuong Hong; Park, Hannah; Singh, Prakarsh; Walker, Dilys

VERSION 1 - REVIEW

REVIEWER	Ana Garces Institute of Nutrition of Central America and Panama
REVIEW RETURNED	28-Aug-2018

GENERAL COMMENTS	This is an interesting paper and research topic. The statistical methods seem to be complicated. Not sure if these could be replicated.
---

REVIEWER	Ashrafal Alam The University of Sydney, Australia
REVIEW RETURNED	19-Sep-2018

GENERAL COMMENTS	The study proposal will potentially benefit maternal and child health & nutrition of a very large number of population by evaluating the impact of using a smartphone app by Anganwadi Workers as a job aid for their routine counseling. 1. The primary outcome (b) need elaboration. How the author define 'appropriate level of counselling'?2. Line 9. It's is not clear if the authors consider quasi-experimental design as a strength or limitation of the study.3. Line 11. Again 'extent or level of counselling' is not adequately clear as a primary outcome. The authors should define this objective clearly.4. The same lack of clarity reappears in lines 34-41. 'Adequate number of counselling' and 'appropriate level of counseling' are vague terms. The proposal requires clear definition of these terms that has specific unit of measurement. I see an absence of this clarity as a major weakness of this protocol.5. Data collection tools are integral element of a research protocol. I suggest the authors attach the tools.
---

REVIEWER	Phillip Phan Johns Hopkins University United States of America
REVIEW RETURNED	10-Oct-2018

GENERAL COMMENTS	The research question asks if an mHealth intervention changes the behavior of the provider. More information is needed to directly connect the intervention to the anticipated change in behavior. If you expect more visits to be the result, then what is it specifically about the intervention (incentives, nudges, gamification, etc.) that is theoretically causing the change the behavior. I'm not sure I understand how you are collecting the data. If you are asking mothers whether they were visited, wouldn't the mothers be incentivised to under-report, so they get more visits/resources? Why not build into the app a 'check-in' mechanism, or even better a GPS-based tracker to objectively collect the data? Ultimately, the intervention is meant to improve maternal and child health. Why not directly measure the psychological and clinical outcomes of the intervention (mortality, morbidity, depression, weight change, and so on)? You spent a great deal of time describing the matching process, which is excellent. How do we know that your sample represents the rest of the population being treated? Good luck on your study.
--

REVIEWER	Yiqiong Xie Associate Editor at BioMed Central Country: United States of America
REVIEW RETURNED	16-Oct-2018

GENERAL COMMENTS	I was invited to review this study protocol with a particular emphasis on the statistical methods and analyses proposed. The analysis plan using mixed effects model seems to be statistically appropriate. My only recommendation is: for the primary outcomes, since there are only a limited number of indicators, it's okay not to correct the p-values for multiple comparisons. However, it seems that for secondary outcomes, there'll be quite a few measures. Therefore, may consider correction of P-values for these secondary outcomes when appropriate.
---

VERSION 1 – AUTHOR RESPONSE

Reviewer 1 Reviewer Name: Ana Garces Institution and Country: Institute of Nutrition of Central America and Panama	
Comment 1: This is an interesting paper and research topic. The statistical methods seem to be complicated. Not sure if these could be replicated.	Response: Thank you for your support. We certainly hope that the analysis is replicable, and in fact, we have planned an internal replication ourselves as is standard practice for

	high impact papers. In the DATA SHARING STATEMENT (page 13) section, we are committing to including questionnaires and an indicator construction sheet along with deidentified data. All models are already prespecified for independent replication. In the ANALYSIS PLAN section (page 11), we have added a sentence “All analyses will be done in STATA, documented in a DO file, and replicated by two different analysts.”
--	---

Reviewer 2 Reviewer Name: Ashrafal Alam Institution and Country: The University of Sydney, Australia	
The study proposal will potentially benefit maternal and child health & nutrition of a very large number of population by evaluating the impact of using a smartphone app by Anganwadi Workers as a job aid for their routine counseling.	Response: Thank you very much for your helpful review.
Comment 1: The primary outcome (b) need elaboration. How the author define 'appropriate level of counselling'?	Response: Thank you for flagging this important omission. We have added more clarification in the PRIMARY OUTCOMES section (page 10) as follows,  1. The proportion of pregnant women and mothers of children <12 months who received adequate number of home visits by the AWW in the past three months (adequate number will be the minimum number of visits a respondent must receive as per the ICDS guidelines for the current life-stage/age. Additionally, we will use, as a supporting indicator, the number of visits as a continuous outcome indicator.[26]); and 2. The proportion of pregnant women and mothers of children <12 months who received appropriate extent or level of counselling from the AWW during their interactions (at home, at AWCs, or in other settings) in the past three months (appropriate level of counselling will be a recall of at least half of the correct messages/counselling that a respondent should receive as per the ICDS guidelines for the current life-stage/age. Additionally, we will use, as a supporting indicator, number of correct messages or services recalled by the respondent as a continuous outcome[26]).
Comment 2: Line 9. It's is not clear if the authors consider quasi-experimental design as a strength or limitation of the study.	Response: We have revised bullet # 2 in STRENGTHS AND WEAKNESSES to be clearer that a cluster RCT is the gold standard and the quasi-experimental method is scientifically less robust. Application of the gold-standard cluster-randomized controlled trial design was not possible because of pre-determined programme assignment and the rapid roll-out of the program.

	Therefore, to find attributable impacts, this evaluation settles for a scientifically less robust but practicable quasi-experimental design consisting of matched control villages and repeated cross-sectional measurements
Comment 3: Line 11. Again 'extent or level of counselling' is not adequately clear as a primary outcome. The authors should define this objective clearly.	Response: Addressed as discussed for comment 1
Comment 4: The same lack of clarity reappears in lines 34-41. 'Adequate number of counselling' and 'appropriate level of counseling' are vague terms. The proposal requires clear definition of these terms that has specific unit of measurement. I see an absence of this clarity as a major weakness of this protocol.	Response: Addressed as discussed for comment 1
Comment 5: Data collection tools are integral element of a research protocol. I suggest the authors attach the tools.	Response: All tools along with the raw data will be made public as already committed in this protocol paper. As a part of the IRB approval docket, the questionnaires will be submitted to the journal where we publish the impact evaluation paper. Therefore, we are not sure if and why tools should also be submitted with the protocol paper. However, if BMJ editorial team so desires, we are willing to submit the baseline questionnaires as supplementary material even for the protocol paper.

Reviewer 3

Reviewer Name: Phillip Phan

Institution and Country: Johns Hopkins University, United States of America

Comment 1: The research question asks if an mHealth intervention changes the behavior of the provider. More information is needed to directly connect the intervention to the anticipated change in behavior. If you expect more visits to be the result, then what is it specifically about the intervention (incentives, nudges, gamification, etc.) that is theoretically causing the change

Response: We felt that the Theory of Change in Figure 2 succinctly summarized how we expect ICDS-CAS to deliver impacts over time. However, we have added the following when we describe the ICDS-CAS App on page 6.

The CAS app is especially expected to improve home visit service delivery by the AWW through improved channels of information (past records of the beneficiary for customized messaging, educational animation videos as a job aid, life-stage-appropriate check-list for counselling messages) and timely nudges (automatic creating of visit due lists, alerts for approaching or missed visits, timely intimation of delays to the lady supervisor). Thus, improved home visits in terms of timeliness, frequency, and perhaps, more effective message delivery mechanism are expected to result in increased knowledge and better recall of correct health and nutrition practices by the beneficiaries and demand for related government services. However, for the actual behaviors to change and sustain, supply side constraints must be

	addressed to meet the demand for services (e.g., adequate supply of supplementary ration, adequate provisions of IFA tablets, regular immunization camps). Such improvements can be expected only in the mid- to long-term because they are beyond the sphere of influence of ICDS-CAS and need more ICDS-wide improvements.
Comment 2: I'm not sure I understand how you are collecting the data. If you are asking mothers whether they were visited, wouldn't the mothers be incentivised to under-report, so they get more visits/resources? Why not build into the app a 'check-in' mechanism, or even better a GPS-based tracker to objectively collect the data?	Response: Thank you for your suggestion. Indeed, we had checked if the back-end database of ICDS-CAS can be shared with the evaluation team to track the App use including interaction of the AWW with beneficiaries. Unfortunately, this data cannot be shared with the study team due to recent data privacy and protection concerns related to government collected data. There are also concerns about the reliability and independence of this data from the evaluation perspective. A recall bias which underestimates the number of visits is likely as is the case with most recall-based subjective questions. There is no real economic incentive to intentionally under-report the number of visits because nothing is “given” except messages during home visits. Therefore, we do not anticipate such a bias to be differential between the intervention and control groups. Second, for the sampled households, we will be collecting actual home visit records from the registers at the AWC. However, data from such registers can be inflated and unreliable, and thus, could over-report the number of visits, which is a more serious error. Therefore, we plan to report this only as a supporting indicator. We will be flagging this as a limitation in the IE manuscript as we have already done in this protocol paper: “Measurement biases may exist because blinding is not possible and primary outcomes are measured subjectively via interview-based recall or observations”
Comment 3: Ultimately, the intervention is meant to improve maternal and child health. Why not directly measure the psychological and clinical outcomes of the intervention (mortality, morbidity, depression, weight change, and so on)?	Response: Thank you for raising this question. As we have clarified in the protocol, psychological and clinical outcomes of the intervention are distal to the evaluation timeframe. This mHealth intervention is just an enabler which can improve service delivery by the AWWs but may not address all supply side constraints (e.g., it cannot fix ration distribution or IFA tablet supply). On the demand side, family support and social norms may not budge in a short follow-up period of <12 months. It is, therefore, unfair to consider health or nutrition as primary outcomes on which the success of the programme will be judged. Therefore, we have consciously used recall of the number of

	visits and counselling received from AWWs as the primary outcomes. As “aspirational” outcomes and for research into the link between mHealth, AWW services, and nutrition/health outcomes, we are also measuring family support, nutritional (growth) outcomes, and delivery outcomes, but we cannot treat them as primary.
Comment 4: You spent a great deal of time describing the matching process, which is excellent. How do we know that your sample represents the rest of the population being treated?	Response: Thank you! We have already clarified as a limitation that “The external validity of the findings can be questioned because of the purposive sampling of states, pairing of districts, and the PSM matching of villages does not result in a statistically representative sample of ICDS-CAS programme area”. At this point, we do not have recent and reliable data on the rest of the ICDS-CAS area in MP or Bihar. We are in talks with ICDS to get annual survey data for all AWCs in MP and Bihar states. If made available, in the endline analysis we will be able to present a table on key population and AWC characteristics for the entire states, ICDS-CAS areas, and the study sample. For now, we must only concede that we cannot claim external validity, which we have done already.

Reviewer 4 Reviewer Name: Yiqiong Xie Institution and Country: Associate Editor at BioMed Central, Country: United States of America	
Comment 1: I was invited to review this study protocol with a particular emphasis on the statistical methods and analyses proposed. The analysis plan using mixed effects model seems to be statistically appropriate. My only recommendation is: for the primary outcomes, since there are only a limited number of indicators, it's okay not to correct the p-values for multiple comparisons. However, it seems that for secondary outcomes, there'll be quite a few measures. Therefore, may consider correction of P-values for these secondary outcomes when appropriate.	Response: Thank you for flagging out this omission. We have added the following sentence when we discuss adjusting p-values in ANALYSIS PLAN on Page 12. However, for secondary outcomes when multiple indicators related to a topic or theme are compared, we will adjust the p-values for multiple comparisons.

VERSION 2 – REVIEW

REVIEWER	Yiqiong Xie Daemen College USA
REVIEW RETURNED	13-Dec-2018

GENERAL COMMENTS	It seems that the authors have addressed reviewer comments. I'd recommend accepting the manuscript.
---